# Inflammatory Signaling and DNA Damage Responses after Local Exposure to an Insoluble Radioactive Microparticle

**DOI:** 10.3390/cancers14041045

**Published:** 2022-02-18

**Authors:** Yusuke Matsuya, Nobuyuki Hamada, Yoshie Yachi, Yukihiko Satou, Masayori Ishikawa, Hiroyuki Date, Tatsuhiko Sato

**Affiliations:** 1Nuclear Science and Engineering Center, Japan Atomic Energy Agency (JAEA), 2-4 Shirakata, Tokai 319-1195, Ibaraki, Japan; sato.tatsuhiko@jaea.go.jp; 2Radiation Safety Unit, Biology and Environmental Chemistry Division, Sustainable System Research Laboratory, Central Research Institute of Electric Power Industry (CRIEPI), 2-11-1 Iwado-kita, Komae 201-8511, Tokyo, Japan; hamada-n@criepi.denken.or.jp; 3Graduate School of Health Sciences, Hokkaido University, Kita-12 Nishi-8, Kita-ku, Sapporo 060-0812, Hokkaido, Japan; yachi1018@frontier.hokudai.ac.jp; 4Collaborative Laboratories for Advanced Decommissioning Science (CLADS), Japan Atomic Energy Agency (JAEA), 790-1 Otsuka, Motooka Tomioka, Futaba 979-1151, Fukushima, Japan; satou.yukihiko@jaea.go.jp; 5Faculty of Health Sciences, Hokkaido University, Kita-12 Nishi-8, Kita-ku, Sapporo 060-0812, Hokkaido, Japan; masayori@med.hokudai.ac.jp (M.I.); date@hs.hokudai.ac.jp (H.D.)

**Keywords:** Cs-BMP, inflammation, DNA damage, cell survival, intercellular communication

## Abstract

**Simple Summary:**

A cesium-bearing microparticle (Cs-BMP) is an insoluble radioactive microparticle possessing high specific radioactivity, which was discovered after the incident at the Fukushima nuclear power plant. Due to their insoluble nature, such Cs-BMPs are assumed to adhere in the long term to normal tissue, leading to chronic local exposure. However, radiation risk due to the intake of internal exposure to radioactive cesium is conventionally estimated from the organ dose given by uniform exposure to soluble cesium. As such, it is critical to clarify the normal tissue effects posed by heterogeneous exposure to Cs-BMPs. This in vitro study reports on the relationship between the inflammatory responses and DNA damage induction during local exposure to a Cs-BMP.

**Abstract:**

Cesium-bearing microparticles (Cs-BMPs) can reach the human respiratory system after inhalation, resulting in chronic local internal exposure. We previously investigated the spatial distribution of DNA damage induced in areas around a Cs-BMP; however, the biological impacts have not been fully clarified due to the limited amount of data. Here, we investigated the inflammatory signaling and DNA damage responses after local exposure to a Cs-BMP in vitro. We used two normal human lung cell lines, i.e., lung fibroblast cells (WI-38) and bronchial epithelial cells (HBEC3-KT). After 24 h exposure to a Cs-BMP, inflammation was evaluated by immunofluorescent staining for nuclear factor κB (NF-κB) p65 and cyclooxygenase 2 (COX-2). The number of DNA double-strand breaks (DSBs) was also detected by means of phospholylated histone H2AX (γ-H2AX) focus formation assay. Cs-BMP exposure significantly increased NF-κB p65 and COX-2 expressions, which were related to the number of γ-H2AX foci in the cell nuclei. Compared to the uniform (external) exposure to ^137^Cs γ-rays, NF-κB tended to be more activated in the cells proximal to the Cs-BMP, while both NF-κB p65 and COX-2 were significantly activated in the distal cells. Experiments with chemical inhibitors for NF-κB p65 and COX-2 suggested the involvement of such inflammatory responses both in the reduced radiosensitivity of the cells proximal to Cs-BMP and the enhanced radiosensitivity of the cells distal from Cs-BMP. The data show that local exposure to Cs-BMP leads to biological effects modified by the NF-κB pathway, suggesting that the radiation risk for Cs-BMP exposure can differ from that estimated based on conventional uniform exposure to normal tissues.

## 1. Introduction

The incident at the Fukushima Daiichi Nuclear Power Plant (FDNPP) took place after the tsunami disaster in 2011, which emitted large quantities of radioactive materials to the environment [1]. In 2013, insoluble, radioactive cesium-bearing microparticles (Cs-BMPs) were discovered from sites near the FDNPP [2]. Such Cs-BMPs can be classified into two types: Type A with a high specific radioactivity and 1–10 μm diameter, and Type B with a low specific radioactivity and 70–400 μm diameter [3,4]. Cs-BMPs are mainly composed of silicate glass [5,6], which may be formed from glass fibers used in heat insulators covering the pipes [6]. Cs-BMPs have been estimated to be widely dispersed over the greater Kanto region (>250 km away from the FDNPP) [7]. Considering that Cs-BMPs are so small in size and light in weight, Cs-BMPs can reach the human respiratory system. From the viewpoint of radiation protection, it is pivotal to investigate the radiobiological impact of Cs-BMPs.

The radiation risk posed by the intake of internal exposure to radioactive cesium is conventionally estimated from the organ dose given by uniform exposure to soluble cesium. Meanwhile, Cs-BMPs are assumed to adhere in the long term to normal tissues due to their insoluble nature. Considering local energy deposition around a Cs-BMP [8], normal tissues are assumed to be chronically and partially exposed at a microenvironmental scale. However, such heterogeneous dose distribution within the tissue has not been considered in evaluating the radiation risk of Cs-BMPs. Considering such gaps, the conventional risk assessment based on the uniform exposure at an organ scale is insufficient. In addition to the estimation of the intra-tissue dose distribution and biokinetics [9,10], the accumulation of the fundamental in vitro data on local exposure is needed for a better understanding of the radiation risk after exposure to Cs-BMPs.

There are several reports on responses of human normal cells to local radiation exposure [8,11,12]. Among them, our previous in vitro study reported on the spatial distribution of DNA double-strand breaks (DSBs), which was detected by the histone H2AX phosphorylated on serine 139 (γ-H2AX foci) at DSB sites [13,14], after 24 h exposure to Cs-BMP with 505.7 Bq [8]. Our previous results suggest that local exposure to a Cs-BMP leads to an increase in DNA damage to distal cells and a decrease in DNA damage to proximal cells. Therefore, the intercellular signaling between irradiated cells and non-irradiated cells (i.e., via non-targeted effects or bystander effects of radiation [15,16,17,18,19]) may modify radiation responses during local exposure. However, other endpoints have not been evaluated, and the underlying mechanisms remain unclear. Therefore, further studies are necessary for clarifying the underlying mechanisms.

Among various radiobiological endpoints, inflammatory responses have been implicated in carcinogenesis [20,21,22,23,24]. It is well known that the inflammatory responses can be quantified by the signaling pathways involving nuclear factor-kappa B (NF-κB) p65 nuclear translocation and cyclooxygenase-2 (COX-2) expression [25]. NF-κB is also a transcription factor that can bind the kappa immunoglobulin-light chain enhancer [26], which triggers the gene expressions responsible for proliferation and anti-apoptosis. The transcription factor activation induces cytokines responsible for immune reactions (such as tumor necrosis factor α (TNF-α) and interleukins IL-1, IL-6 and IL-8). Due to the adhesion molecules that attract leukocytes to the inflammation sites, it is interpreted that dysregulation of the transcription factor is related to chronic inflammatory diseases and cancer development. Meanwhile, COX-2 is a secondary signaling molecule (downstream of NF-κB [25]) that is produced after stimulation with cytokines and mitogens [27], which is known to be involved in inflammation processes. As inflammatory responses are mediated by NF-κB p65 and COX-2 pathways, the evaluation of the role of these signaling pathways is of importance in discussing radiation effects after inhaling Cs-BMPs. Here, we set out to investigate the inflammatory signaling pathways after Cs-BMP exposure, and evaluate the relationship between inflammatory response and DSB induction, in comparison to responses following uniform exposure to ^137^Cs γ-rays.

## 2. Materials and Methods

### 2.1. Cell Culture

Considering that a Cs-BMP can adhere to lung tissue, we used two types of normal human diploid lung cell lines: WI-38 primary lung fibroblasts (RCB0702, RIKEN, Tokyo, Japan) and HBEC3-KT bronchial epithelial cells immortalized with hTERT and CDK4 (CRL-4051, ATCC, Manassas, VA, USA). WI-38 cells were maintained in Dulbecco’s modified Eagle’s medium/Nutrient Mixture F-12 (DMEM/F12) (D8437, Sigma, Kawasaki, Japan) supplemented with 10% fetal bovine serum (FBS, Nichirei Bioscience Inc., Tokyo, Japan), and the HBEC3-KT cells were maintained in bronchial epithelial cell medium (3211NZ, ScienCell, Carlsbad, CA, USA), as described in [8]. WI-38 and HBEC3-KT cells were maintained at 37 °C in a humidified atmosphere of 5% CO_2_ in air. 

### 2.2. Irradiation Setups

Cells seeded on the glass-based dish with φ35 mm (3911-035, IWAKI, Chiyoda, Japan) were continuously exposed to the Cs-BMP (the sample ID: CF-01 [4]) for 24 h using the microcapillary (MP-020, MicroSupport Co., Ltd., Shizuoka, Tokyo), as described in [8]. In our previous study [8], there was no significant difference between 24 h exposure and 48 h exposure, so we employed 24 h exposure in this study as well. The Cs-BMP used in this study was classified as a Type B particle composed of 94.5% ^137^Cs with 455.2 Bq and 5.5% ^134^Cs with 26.5 Bq as of 13 January 2020. The irradiation experiments were repeatedly performed at several times during the period from 13 January 2020 to 29 July 2021. Figure 1A illustrates the experimental geometry using the Cs-BMP enclosed in the microcapillary. Figure 1B is the dose profile around the Cs-BMP obtained using the Particle and Heavy Ion Transport code System (PHITS) version 3.08 [28], adapting the electron gamma shower (EGS) mode [29] and RI source database (ICRP07) [30]. Note that the dose profile was evaluated for each irradiation experiment. The cutoff energies for photons and electrons were set as 1 keV. Based on the PHITS calculation, the mean boundary between the β-ray dominant area and γ-ray dominant area was found to be 1650 μm, as described in [8].

The homogeneous exposure of the entire area of the cell culture dish (hereafter called uniform exposure) for 24 h to ^137^Cs γ-rays was also performed to compare the local exposure to the Cs-BMP. The absorbed dose rates used in this study were 0.450, 0.100, 0.050, 0.010, 0.005 and 0.001 Gy/day. The dose rates were obtained from the measurement using the Farmer-type ionizing radiation chamber (model NE2581, Nuclear Enterprises Ltd., Reading, UK) based on the International Atomic Energy Agency (IAEA) Technical Report Series No. 277 [31] and were validated by the PHITS calculation. 

In addition, we also performed a non-uniform exposure to the cell culture dish using 6-MV X-rays from Clinac 600EX, in which the 50% cells were exposed by the placement of cell culture container at the edge of the radiation field. The field size was 10 cm × 10 cm, and the water equivalent depth from the phantom surface was 10 cm. The in-field and out-of-field dose rates were measured using a Farmer-type ionization chamber, which was calibrated according to Japanese Standard Dosimetry 1 [32], and the detailed relative dose profile was obtained by Gafchromic EBT3 film and the PHITS calculation using the phase-space file for Varian Clinac 600C (equivalent to Clinac 6EX) 6MV photons. The geometry and normalized dose profile are depicted in Appendix A, in which the PHITS calculation agreed well with both measured values.

### 2.3. Immunofluorescent Staining for NF-κB p65 and COX-2

Following the 24 h exposure, the cells were fixed immediately by 4% paraformaldehyde (PFA) for 10 min on ice. After rinses with phosphate buffered saline (PBS), the cells were permeabilized in 0.2% Triton X-100 in PBS for 5 min, and were then blocked in 1% bovine serum albumin (BSA) in PBS for 1 h. The cells were incubated at 4 °C overnight with primary antibodies against NF-κB p65 (200-301-065, Rocklands, Washington, DC, USA) and COX-2 (ab52237, Abcam, Cambridge, UK) diluted at 1:250 by the 1% BSA in PBS. After rinses with the 1% BSA in PBS three times, the cells were incubated for 2 h with Alexa Fluor 594-conjugated goat-anti-mouse IgG H&L (ab150116, Abcam, Cambridge, UK) and Alexa Fluor 488-conjugated goat-anti-rabbit IgG H&L (ab150077, Abcam, Cambridge, UK) diluted at 1:250 by the 1% BSA in PBS. After rinses with the 1% BSA in PBS three times, the cells were incubated with 1 μg/mL DAPI (62248, Thermo Fisher Scientific, Waltham, MA, USA) for 15 min. After rinses with methanol once, NF-κB p65 nuclear translocation and COX-2 expression were observed using a Keyence BZ-9000 fluorescent microscope (Osaka, Japan).

Representative images of the immunofluorescent staining for NF-κB p65 and COX-2 after 24 h exposure to Cs-BMP are shown in Figure 2A, where left, central and right images represent the sham-irradiated group, β-ray dominant area (*R* ≤ 1650 μm) and γ-ray dominant area (1650 μm < *R*), respectively. NF-κB and COX-2 positive cells were evaluated by both manual assessment and the fluorescent intensity threshold. In Figure 2A, blue, red and green fluorescent intensities represent the cell nuclei stained by DAPI, NF-κB p65 and COX-2, respectively. Note that yellow fluorescent intensity denotes double-positive cells.

### 2.4. γ-H2AX Focus Formation Assay

Prior to the Cs-BMP exposure, the cells were treated overnight with 1 μM BAY 11-7082 (AG-CR1-0013-M010, AdipoGen, San Diego, CA, USA) as an inhibitor of the NF-κB nuclear translocation, 50 μM NS-398 (70590, Cayman Chemical, Ann Arbor, MI, USA) as the COX-2 inhibitor, or both. As these inhibitors were solved in dimethylsulfoxide (DMSO), we included a mock-treated group in which cells were treated with 0.04% DMSO and exposed to the Cs-BMP for 24 h. After exposure, the cells were fixed by 4% PFA for 10 min on ice, rinsed, and permeabilized in 0.2% Triton X-100 in PBS for 5 min. The cells were then blocked in 1% BSA in PBS for 30 min, and were incubated at 4 °C overnight with primary antibodies against γ-H2AX (ab26350, Abcam, Cambridge, UK) diluted 1:400 by the 1% BSA in PBS. After rinsing with the 1% BSA in PBS three times, the cells were incubated for 2 h with Alexa Fluor 594-conjugated goat-anti-mouse IgG H&L (ab150116, Abcam, Cambridge, UK) diluted by 1:250 by the 1% BSA in PBS. After rinses with the 1% BSA in PBS three times, the cells were incubated with 1 μg/mL DAPI (62248, Thermo Fisher Scientific, Waltham, MA, USA) for 15 min. After rinses with methanol once, γ-H2AX foci were observed under the Keyence BZ-9000 fluorescent microscope. 

Representative images of the γ-H2AX foci after 24 h exposure to Cs-BMP are shown in Figure 2B, in which left, central and right images represent the sham-irradiated group, β-ray dominant area (*R* ≤ 1650 μm) and γ-ray dominant area (1650 μm < *R*), respectively. The green signal inside the cell nucleus (blue area) represents γ-H2AX focus. To obtain the spatial distribution of nuclear γ-H2AX foci around the Cs-BMP, γ-H2AX focus was evaluated with automated foci count (peak search method) using the ImageJ software [33,34], as described in [8].

### 2.5. Clonogenic Assay

The cultured cells were counted by using a hemocytometer (Erma, Tokyo, Japan) and plated in T25 flasks (156367, Nunc, Waltham, MA, USA). The cells were allowed to adhere overnight prior to irradiation. After exposure to 6-MV-lianc X-rays (Clinac 6EX, Varian, Palo Alto, CA, USA), cells were incubated for 14 days at 37 °C in a humidified atmosphere of 95% air 5% CO_2_. Colonies were fixed with methanol and stained with 2% Giemsa solution (Kanto Chemical Co. Inc., Tokyo, USA). When calculating surviving fraction, the colonies located in the penumbra regions (−1.0 < *x* [cm] < 1.0 in Appendix A) were excluded. The surviving fraction is the ratio of plating efficiency of the irradiated group to that of the non-irradiated group. The PHITS calculation showed that the out-of-field dose relative to the in-field dose is 5.0% on average (Appendix A).

### 2.6. Statistics

The significant differences among mean values in inflammatory responses (NF-κB p65 nuclear translocation and COX-2 expression) were evaluated by a multiple comparison method, the Tukey–Kramer test. Meanwhile, because the nuclear γ-H2AX foci did not follow the normal distribution, and there was no homoscedasticity between the sham-irradiated group and irradiated group, the significant differences of nuclear foci in number were evaluated by using a multiple comparison method, the Scheffe’s F test. Based on the statistical indices, we evaluated the impact of local Cs-BMP exposure on inflammatory signaling and DNA damage responses. To evaluate the impact of non-uniform exposure on clonogenic survival, we used a paired *t*-test. Based on the p-value, we evaluated the impact of non-uniform exposure on cell survival, compared to the uniform-field exposure.

## 3. Results and Discussion

### 3.1. Spatial Distribution of Inflammation under a Cs-BMP Exposure

We measured the spatial distribution of the inflammatory responses around a Cs-BMP. Figure 3 shows the fractions of the NF-κB-positive, COX-2-positive and double-positive cells after 24 h exposure to the Cs-BMP, in WI-38 cells (Figure 3A) and HBEC3-KT cells (Figure 3B). The boundary between the β-ray and γ-ray dominant areas was 1650 μm. Here, we compared four groups, i.e., non-irradiated cells, all exposed cells at a radial distance *R* ≤ 3300 μm, the proximal cells at *R* ≤ 1650 μm and the distal cells at 1650 < *R* ≤ 3300 μm. There was a significant increase in the NF-κB p65 nuclear translocation in both cell lines located within a 3300 μm distance from the Cs-BMP. The dual activation of NF-κB and COX-2 was observed in the proximal region (≤1650 μm) for WI-38 cells and in the entire region for HBEC3-KT cells. The pattern of the dual activation for NF-κB and COX-2 resembles that of nuclear foci of γ-H2AX after the Cs-BMP exposure reported in our previous paper [8].

As shown in Figure 1B, the energy deposited within cells gradually decreases with increasing radial distance from a Cs-BMP. In this experiment, the distance giving 1 mGy, at which a human fibroblast nucleus is traversed on average by approximately one electron track [35], was ~1550 μm. In this regard, the cells at >1550 μm can be categorized as bystander cells (non-hit cells). The bystander effects (intercellular signaling) can induce not only DNA damage [36,37,38] but also inflammatory responses [25,39,40] in non-hit cells. Therefore, the significant dual activation can predominantly be due to the bystander effects. In general, the diffusion distance of intercellular signals is 90–100 μm for calcium [41] and < 800 μm for apoptosis induction [42]. However, the range seems to be much longer in this experimental system.

### 3.2. Comparison of Inflammation between Cs-BMP and ^137^Cs γ-rays

The fractions of the NF-κB-positive, COX-2-positive and double-positive cells after the Cs-BMP exposure were next compared to those after the uniform exposure to ^137^Cs γ-rays. Figure 4 shows a comparison between local exposure to the Cs-BMP and uniform exposure to ^137^Cs γ-rays, where Figure 4A,B show NF-κB p65 nuclear translocation, Figure 4C,D show COX-2 and Figure 4E,F show dual activation. From the left bar graph in each panel, there was no difference between sham-irradiated groups for both irradiation regimens in both cell lines. The local exposure to Cs-BMP could induce significant activation of NF-κB p65 in the wide dose range (Figure 4A,B). In particular, the cells closest to the Cs-BMP tended to show increased translocation of NF-κB p65, compared to those after uniform exposure. Meanwhile, the activation of the COX-2 singling pathway and the double pathways tended to increase in a low-dose range below approximately 0.05 Gy (Figure 4E,F), which intriguingly is in accordance with the value of γ-H2AX focus formation reported in our previous study [8].

We evaluated the relationship between the increase in double-positive cells and the number of radiation-induced γ-H2AX foci. Figure 5A shows the relation for Cs-BMP exposure and Figure 5B shows that for ^137^Cs γ-ray exposure (the data on γ-H2AX foci taken from our previous report [8]). As shown in Figure 5, the inflammatory signaling pathways are related to the nuclear number of γ-H2AX foci (DSBs). Ataxia telangiectasia mutated and the NF-κB modulator cooperate to activate the NF-κB pathway in response to DNA damage [43]. Additionally, the cytokines, such as IL-6 and IL-8, result in the secondary activation of NF-κB and COX-2 pathways, inducing DNA damage in bystander cells [25,44,45]. Thus, a correlation between radiation-induced γ-H2AX foci and an increase in the double-activation seems to be reasonable. However, the R^2^ value for Cs-BMP exposure is lower than that for uniform exposure (see Figure 5), due to DNA damage responses modified by the NF-κB pathway. Lam et al. reported higher activation of phosphorylated NF-κB when 2.5% of the cell population was irradiated than when 100% was irradiated, which is involved in rescue effect induction (reduction of 53BP1 foci) [46]. Considering these, NF-κB and COX-2 pathways might play an important role in modifying the DNA damage responses under Cs-BMP exposure. However, we separately performed immunofluorescent staining for inflammatory signaling expression and γ-H2AX focus formation assay due to limited function of the fluorescent microscopy used in this study. In future, the triple staining technique is expected to enable a clear correlation between DNA damage and inflammation signaling to be obtained.

### 3.3. DSB Induction when Inflammatory Signaling Is Inhibited

To further investigate the relationship between inflammatory signaling and DNA damage response, we used 1 μM Bay 11-7082 and 50 μM NS-398 as inhibitors of NF-κB p65 and COX-2 pathways, respectively [39,47]. The effectiveness of these inhibitor concentrations was validated, as shown in Appendix A, showing that NF-κB p65 and COX-2 activations decline to the non-irradiated level for both WI-38 and HBEC3-KT cells. At these concentrations, we measured nuclear γ-H2AX foci as a function of the radial distance from the Cs-BMP for various inhibitor treatments.

Figure 6 shows the nuclear γ-H2AX foci after the Cs-BMP exposure for various treatments with inflammatory pathway inhibitors, in which the upper panel (Figure 6A–C) shows the WI-38 cell line and the lower panel (Figure 6D–F) shows the HBEC3-KT cell line. The spatial distributions of nuclear γ-H2AX foci for various treatments are summarized in Appendix A, from which two areas were selected: one is closest to Cs-BMP (i.e., at 0–300 μm), and the other is the boundary between the β-ray dominant and γ-ray dominant areas (1650–1950 μm). The left panel shows the sham-irradiated cells, the middle panel shows the cells located at 0–300 μm from the Cs-BMP and the right panel shows the cells located at 1650–1950 μm from the Cs-BMP. In the sham-irradiated WI-38 cells (Figure 6A), there were no significant difference among the sham-irradiated groups, indicating the lack of impact of these inhibitors on DNA damage induction for WI-38 cells. Meanwhile, these inhibitors could affect the number of γ-H2AX foci in sham-irradiated HBEC3-KT cells (Figure 6B). From these results, we calculated the γ-H2AX foci relative to the sham-irradiated group, and evaluated the impact of the inhibition of NF-κB and COX-2 on DNA damage induction after Cs-BMP exposure. Figure 6F shows that the inhibition of NF-κB and COX-2 reduced DNA damage (DSB) induction in the HBEC3-KT cells at 1650–1950 μm, suggesting that DNA damage induction under low-dose-rate exposure is involved in the activation of the inflammatory signaling pathway by bystander effects [44,48,49]. In the cells at 0–300 μm, the COX-2 suppression also reduced DSB induction (Figure 6B,E); however, the NF-κB p65 suppression elicited the opposite effects (enhancing DSB induction). In support of this, NF-κB inhibition induced detrimental effects on the 53BP1 foci increase [50] and differentiation (culture phenotype) [51].

NF-κB inhibition downregulates COX-2 expression [25,51], thereby mitigating inflammation (Figure 6C,F). In particular, the subsequent COX-2 expression positively modulates bystander cells [44]. Additionally, the in vitro study using aminoguanidine showed that inducible nitric oxide synthase (iNOS) plays a similar role in inducing bystander effects on DSB induction [38]. Reactive oxygen species (ROS) was identified as the messenger of DNA damage induction by bystander effects [8]. Meanwhile, the NF-κB pathway not only stimulates DSB repair, particularly homologous recombination (HR) [52,53,54], but also is associated with a decrease in the ROS level in irradiated cells [55,56]. The NF-κB signaling pathway has been attributed to an inducer for radioresistance due to its antiapoptotic function [57]. Our previous report suggested that γ-H2AX focus formation was less manifested in the cells proximal to the Cs-BMP, which was interpreted by the reduced yield of early damage (so called protective effects) [8]. Nitric oxide-mediated bystander effects can be a trigger to cause radioresistance [58]; however, the inhibition of iNOS did not diminish protective effects [59,60]. Therefore, it is suggested that the upregulation by NK-κB plays a key role in the protective effects.

### 3.4. Cell Survival after Heterogeneous Exposure to X-rays

The NF-κB pathway also negatively controls apoptosis, i.e., towards cell survival [25,57]. To further discuss the normal tissue effects of heterogeneous exposure, the surviving fractions of WI-38 cells and HBEC3-KT cells were measured using the non-uniform exposure technique [59,60]. Figure 7 shows the surviving fraction measured by a clonogenic assay, where the green symbol shows the survival after uniform exposure, and the blue and red symbols show the survival of in-field cells and out-of-field cells, respectively. As shown in Figure 7, the in-field cells under the non-uniform exposure showed lower radiosensitivity than those under uniform-field exposure (indicating protective effects), which showed a tendency similar to the observation in AG01522 normal human foreskin fibroblasts [59]. The HBEC3-KT cells exhibited more protective effects than the WI-38 cells. Meanwhile, the out-of-field cells under the non-uniform exposure showed higher radiosensitivity than those under the uniform-field exposure at the same absorbed dose. Note that on average, the out-of-field dose was 5% of the in-field dose. In the same manner as for the protective effects, HBEC3-KT cells were radiosensitive to intercellular signaling, leading to more cell killing compared to WI-38 cells.

The out-of-field cells exhibited enhanced radiosensitivity due to intercellular signaling from in-field cells to out-of-field cells, such as via NO and ROS [59,60]. Several reports have shown that the inhibition of iNOS (an upstream of NO) by aminoguanidinecan and inhibition of ROS by DMSO can rescue the enhanced radiosensitivity of out-of-field cells [37,38,61]. We also detected the significant activation of NF-κB and COX-2 in out-of-field cells at 2 h after non-uniform exposure to 4 Gy (Appendix A) as bystander responses [25,44,48]. The persistent COX-2 expression was also detected at 16 h after exposure (Appendix A). Therefore, the death of out-of-field cells is attributed to inflammatory responses via intercellular signaling. By changing the irradiation area of culture flask, i.e., 25%, 50%, 75% and 100%, we also measured in-field and out-of-field cell survival (Appendix A). The bystander effects in out-of-field cells seemed to be saturated for a larger field size than 50%. Meanwhile, the protective effects were maximal at the smallest in-field size of 25% (Appendix A). Ojima et al. reported the increase in nuclear 53BP1 foci with X-irradiated field size [12], and Lam et al. reported rescue effects on 53BP1 foci at 12 h post-irradiation with a 2.5% irradiated cell population [46]. The tendency of survival data (Appendix A) agrees well with these data on 53BP1 foci [12,46]. The potential underlying mechanisms of protective effects might be attributable to the reduction in early DNA damage by antioxidants [8,59] or stimulated DNA repair [52,53,54]. The positive effects for irradiated cells might result from NF-κB/Rel and inhibitor of NF-κB (IκB) gene families as mediators of immune responses [44,62] by intercellular communication from irradiated to non-irradiated cells and vice versa (so called bystander cross-talk) [63]. However, due to the limited amount of experimental data, the underlying mechanisms on inducing protective effects remain uncertain, warranting further experiments.

### 3.5. A Scenario Model of the Signaling Pathways under Cs-BMP Exposure

Based on our in vitro experimental results, we modeled scenarios on signaling pathways regulating radiobiological effects modulated by intercellular signaling under Cs-BMP exposure. Figure 8 shows the relationship among inflammatory signaling responses, DSB induction and cell death during exposure to Cs-BMP (non-uniform exposure). Figure 8A is the schematic illustration of Cs-BMP exposure and dose of Cs-BMP. Figure 8B is a hypothetical model of the signaling pathways regulating radiobiological effects under Cs-BMP exposure.

The energy locally deposited in proximity to a Cs-BMP, and the boundary between the β-ray dominant area and γ-ray dominant area, was 1650 μm on average (Figure 1B). During continuous local exposure to Cs-BMP, β-rays and secondary electrons induced DNA damage by energy deposition [64] in mainly proximal cells by targeted effects to directly irradiated cells. Such DNA damage can be repaired by non-homologous end joining (NHEJ) and homologous recombination (HR) [65]. DNA damage by targeted effects activated the NF-κB pathway and downstream COX-2 expression, while intercellular signals induced non-targeted effects and also activated the inflammatory signaling pathway (NF-κB and COX-2) (see Figure 3 and Figure 4), as well as iNOS expression [59,60]. Considering ROS-mediated DNA damage induction in non-hit cells in our previous study [8], the inflammatory signaling finally generated ROS and NO in non-hit cells (distal cells) and induced DNA damage through bystander effects [8,59], leading to enhanced cell death induction (see Figure 7) [59]. Meanwhile, the NF-κB pathway suppressed the ROS level using antioxidants and reduced DNA damage induction (see Figure 6B,E) [46,55,56,59] (DNA repair stimuli [52,53,54]), and controlled antiapoptic function in irradiated cells (Figure 7) [57,59]. Altogether, intercellular signaling induced protective effects in irradiated (hit) cells and bystander effects in non-irradiated (non-hit) cells, respectively. 

## 4. Conclusions

Assuming that Cs-BMP adheres in the long term to the lung, we investigated the relationship between the inflammatory signaling and DNA damage responses during continuous local exposure to a Cs-BMP (mimicking localized internal exposure). We observed the significant activations of NF-κB p65 and COX-2 after 24 h exposure to Cs-BMP. Such inflammatory signaling pathways were related to DNA damage induction (evaluated as γ-H2AX focus formation). Compared with responses to uniform exposure to ^137^Cs γ-rays (mimicking external exposure), we found that NF-κB signaling plays an important role in protective effects on the irradiated (hit) cells proximal to Cs-BMP and the bystander effects on the non-irradiated (non-hit) cells distal from Cs-BMP. This work shows that the local exposure to Cs-BMP induces different biological effects modified by inflammatory signaling responses from the conventional effects of uniform exposure.

The radiation risk posed by internal exposure due to the intake of radioactive cesium is conventionally estimated based on uniform exposure to soluble cesium at an organ scale. The present in vitro data suggest that the risk assessment based on uniform exposure may not apply for continuous local exposure to Cs-BMP. However, the scientific data reporting on radiobiological impacts under non-uniform exposure are limited. In particular, in vitro experiments focusing on mutation frequency (or direct observation of carcinogenesis) or using 3D culture models, as well as in vivo studies, are essential for a more precise understanding of radiobiological impacts under heterogeneous exposure. Additionally, mechanistic studies using biophysical models will offer an effective approach to mechanistically interpreting the scenario of intercellular communication under continuous heterogeneous exposure.

## Figures and Tables

**Figure 1 cancers-14-01045-f001:**
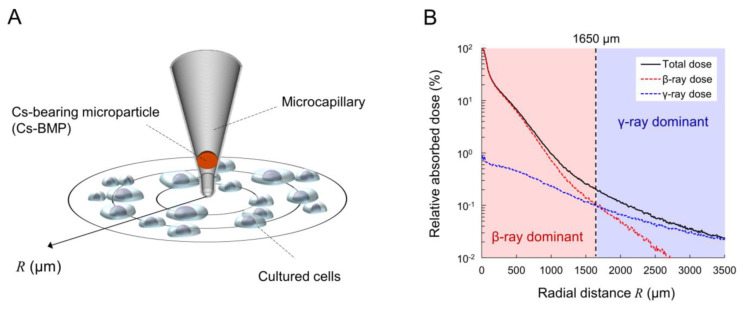
Experimental setup for the Cs-BMP exposure: (**A**) Geometry. (**B**) Dose profile. The dose was calculated by a Monte Carlo-based PHITS code [28]. The boundary between β-ray dominant area and γ-ray dominant area was found to be 1650 μm, as described in [8].

**Figure 2 cancers-14-01045-f002:**
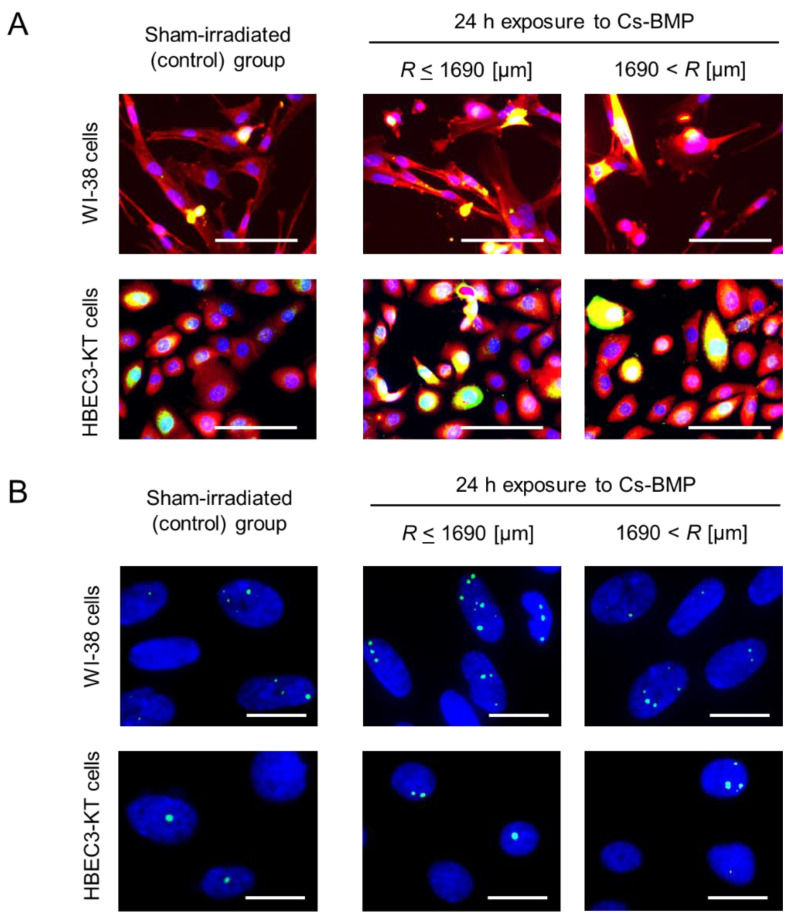
Representative images of the immunofluorescent staining after 24 h exposure to a Cs-BMP. (**A**) The expression of NF-κB p65 and COX-2; (**B**) the γ-H2AX focus. In this figure, left, central and right images represent sham-irradiated group, β-ray dominant area (*R* < 1650 μm) and γ-ray dominant area (1650 μm < *R*), respectively. In (**A**), blue, red and green fluorescent intensities represent cell nuclei stained by DAPI, NF-κB p65 and COX-2, respectively. Note that yellow fluorescent intensity denotes double-positive cells. In (**B**), green signal inside cell nucleus (blue area) represents γ-H2AX focus. Scale bars for (**A**) and for (**B**) are 100 and 20 μm, respectively.

**Figure 3 cancers-14-01045-f003:**
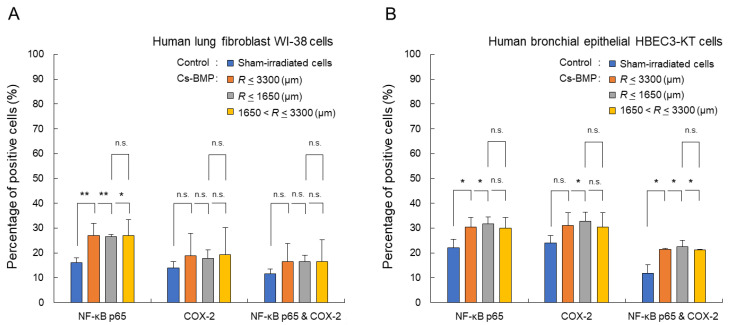
Fractions of the NF-κB p65-positive, COX-2-positive and double-positive cells after 24 h exposure to the Cs-BMP. (**A**) WI-38 cells. (**B**) HBEC3-KT cells. The significant increase was evaluated for three groups, i.e., whole cells at *R* ≤ 3300 μm, the proximal cells at *R* ≤ 1650 μm and the distal cells at 1650 μm < *R* ≤ 3300 μm. The error bar means the standard error of the mean (s.e.m.). The symbols (*, **, n.s.) indicate the 5%, 1% significant difference and non-significant, respectively.

**Figure 4 cancers-14-01045-f004:**
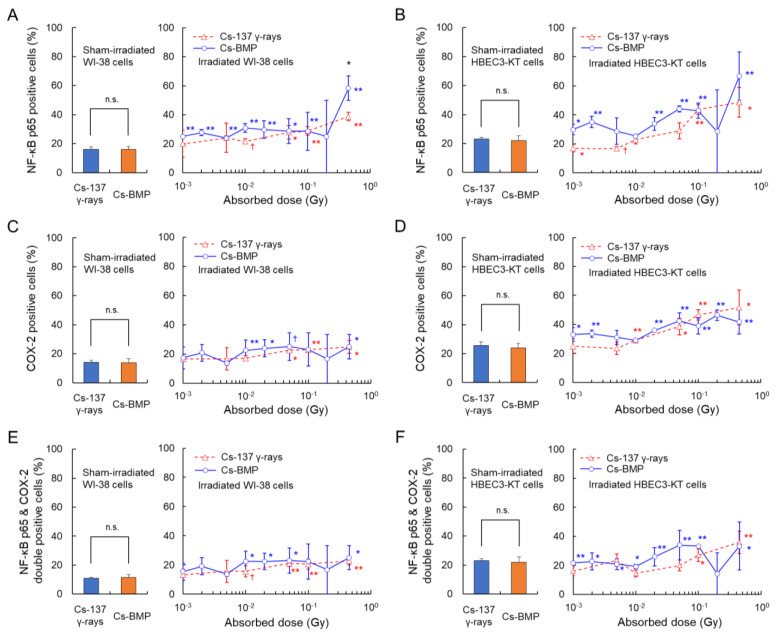
Comparison between local exposure to the Cs-BMP and uniform exposure to ^137^Cs γ-rays. Panels (**A**,**B**), NF-κB p65 nuclear translocation. Panels (**C**,**D**), COX-2. Panels (**E**,**F**), dual activation. Panels (**A**,**C**,**E**), WI-38 cells. Panels (**B**,**D**,**E**), HBEC3-KT cells. In each panel, the left bar graph shows the levels at 0 Gy. The blue and red symbols (*, **) indicate the 5% and 1% significant difference, respectively, compared to sham-irradiated groups. The error bar means the standard error of the mean (s.e.m.). The black symbols represent significant difference between local exposure to the Cs-BMP and uniform exposure to ^137^Cs γ-rays. n.s., non-significant.

**Figure 5 cancers-14-01045-f005:**
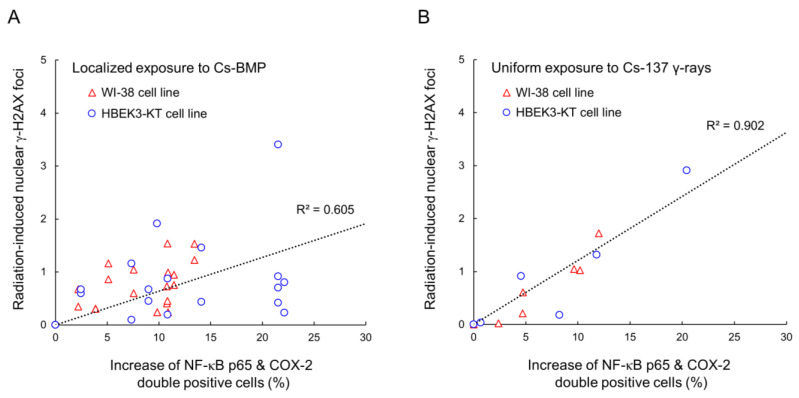
Relationship between the increase of double-positive cells and the number of radiation-induced γ-H2AX foci: (**A**) for the Cs-BMP exposure and (**B**) the ^137^Cs γ-rays exposure. The dotted line represents the linear regression model. The strong correlation between radiation-induced γ-H2AX foci and increase of the double-activation of inflammatory signaling is shown. To discuss the trend using the sufficient amount of experimental data, we used the data on γ-H2AX foci in our previous report [8].

**Figure 6 cancers-14-01045-f006:**
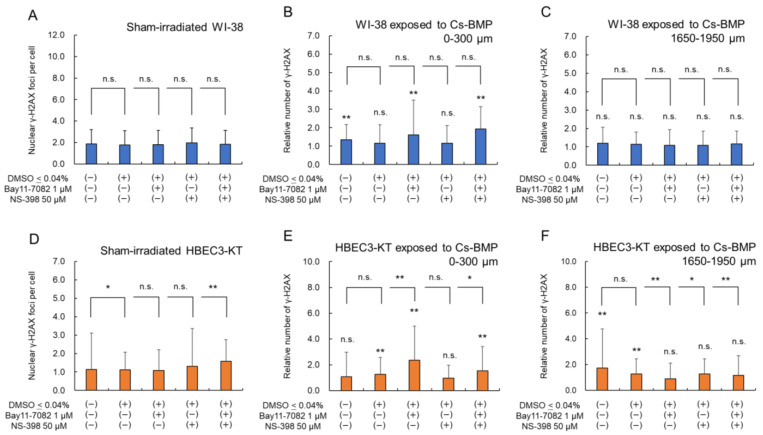
Nuclear γ-H2AX foci after the Cs-BMP exposure for various treatments with inflammatory pathway inhibitors. The upper panel (**A**) shows WI-38 cell line and the lower panel (**B**) shows HBEC3-KT cell line. The left panels show the sham-irradiated cells. The middle panels are the cells proximal to the Cs-BMP within 300 μm. The right panels are the cells distal to the Cs-BMP in the region from 1650 to 1950 μm. Both cell lines were exposed to the Cs-BMP for 24 h. The vertical axis in panels (**A**,**D**) shows the number of γ-H2AX foci per cell, while those in (**B**,**C**,**E**,**F**) show the relative number of γ-H2AX foci to sham-irradiated group. The error bar means the standard deviation (s.d.). The symbols (*, **) indicate the 5% and 1% significant difference, respectively, while the value represents the p-value. n.s., non-significant. The symbol on each bar graph indicates the significance or non-significance compared to sham-irradiated group.

**Figure 7 cancers-14-01045-f007:**
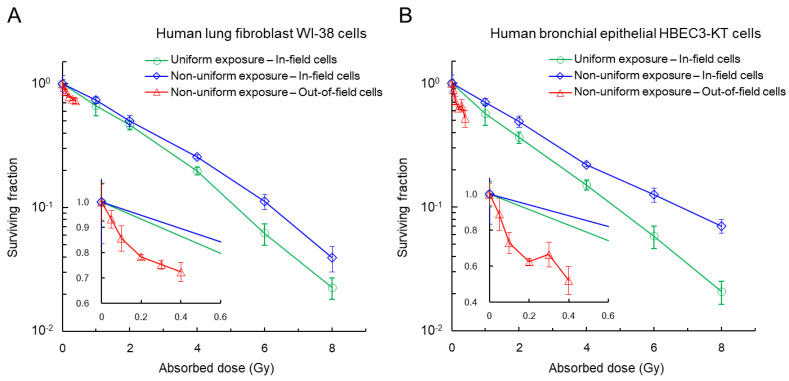
Clonogenic survival: (**A**) WI-38. (**B**) HBEC3-KT. Green symbol, the survival after uniform exposure. Blue, in-field cells. Red, out-of-field cells. There were significant differences between in-field survival after non-uniform exposure and that after uniform-field exposure, except for 1 Gy and 2 Gy in WI-38 cells. The inset represents the low-dose range. The out-of-field dose is 5%, on average, of in-field dose.

**Figure 8 cancers-14-01045-f008:**
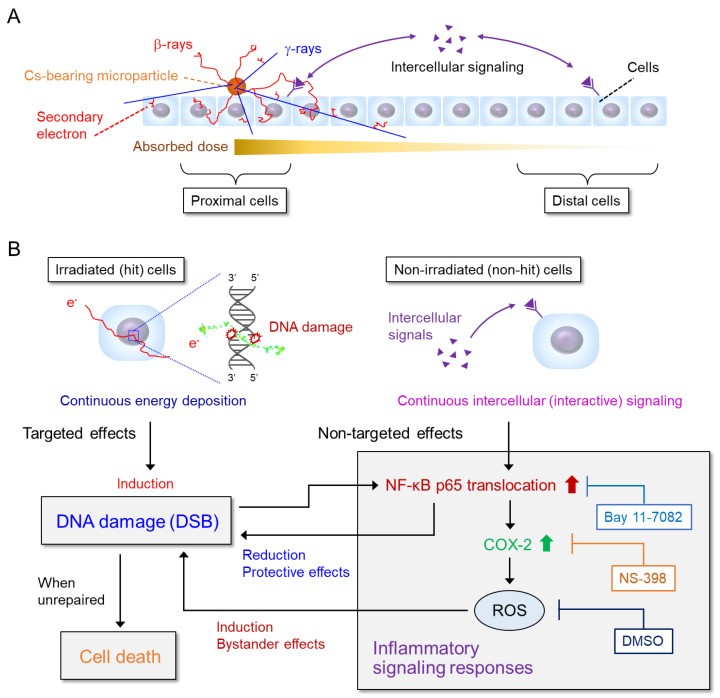
Relationship among inflammatory signaling responses, DNA damage (DSB) induction and cell death under continuous local exposure to Cs-BMP (non-uniform exposure). (**A**) Illustration of radiation emitted from Cs-BMP (β-rays, γ-rays and secondary electrons) and absorbed dose around Cs-BMP. (**B**) A hypothetical model of the signaling pathways regulating radiation effects modulated by intercellular signaling under heterogeneous exposure (bystander effects and protective effects).

## Data Availability

All data generated or analyzed during this study are included in this published article and its supplementary information files.

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
