# Peer review of "Inflammatory Signaling and DNA Damage Responses after Local Exposure to an Insoluble Radioactive Microparticle"

_cancers, 2022, doi:10.3390/cancers14041045_

Round 1
Reviewer 1 Report
The authors presented an in vitro study entitle “Inflammatory Signaling and DNA Damage Responses after Local Exposure to an Insoluble Radioactive Microparticle”, which reports some interesting results on the relationship between the inflammatory responses and DNA damage induction during chronic local exposure to cesium-bearing microparticle (Cs-BMP) discovered after the incident at the Fukushima nuclear power plant.
General comment
The research design, methodology, data comparison as well as statistical analyses performed are appropriate.
The results of this study also provides insightful perspectives for future research on the investigation of the effects of radiation on organisms.
This reviewer particularly appreciated Scinario Model of the Signaling Pathways under Cs-BMP Exposure section and related figure 7. However, the authors did not include any measurement of oxidative stress. Although COX-2 is well-known NF-κB target involved in ROS mediated inflammation pathway, and the authors mention ROS as primary messenger of DNA damage induction, quantification of the intracellular percentage of ROS by fluorimetric assays should be considered to to give more emphasis to the involvement of ROS in DNA damage in irradiated lung fibroblast cells and bronchial epithelial cells.
Specific comment
Why were the cell cultures exposed for only 24 hours? different exposure times would give a clearer view of the DNA repair mechanisms involved.
A representative images of the Immunofluorescent Staining for NF-κB p65 and COX-2 should be included.
The authors should consider to addressed the issues mentioned above before acceptance.
Author Response
We would like to thank the reviewer #1 for careful reading. Please see the attachment.

Reviewer 2 Report
Matsuya and collages investigated a relationship between cellular inflammatory response and DNA damage evoked by a local exposure to a cesium-bearing microparticles (Cs-BMP). Understanding the biological effects of such exposure is critical for assessing ionizing radiation-associated health risks, especially in the light of catastrophic incidents at nuclear power plants. The study finds, that cellular response to the Cs-BMP exposure vary, depending on how far from the radiation source cell is present, and ultimately highlights an involvement of NF-κB and COX-2 signaling in the observed and reported radiation effects. The Authors tackled study hypothesis with an interesting and well-designed experimental setup. The study setup utilized a relevant in vitro model, composed of two normal human lung cell lines, and irradiation systems, which included three modes of radiation exposure (local Cs-BMP, homogenous Cs γ-rays exposure, and non-uniform X-rays exposures). Generally, the Authors are to be applauded for their efforts to precisely describe the experimental setup, dosimetry, and comparison between different radiation exposures modes. However, the study can be improved in a few points, which in turn may lead to strengthening of the conclusions and better understanding of the biological aspects of the work.
Major points:
- The study is mainly based on immunofluorescent (IF) staining of γH2A.X, NF-κB, and COX-2. The readers might miss the exact criteria, on basis of which cells were classified as positive (manual assessment, mean fluorescent intensity threshold or automated foci count?). In addition, if fluorescent microscopy is a method of choice, exemplary immunofluorescence images generally will be a good complement to the figure.
- DNA damage response is composed of rapid processes and fluctuations in signaling pathways depending, i.e., on dose, the duration of exposure, and time after exposure to DNA damaging agent. The Authors focus mainly on reporting the effects of 24h exposure to Cs-BMP, describing it as a ‘chronic exposure’, however the manuscript is missing an explanation of the rationale of this time-point choice. Are the effects of exposure to Cs-BMP in regard to inflammatory signaling peaking at that time-point? Are those effects having a similar (onset) kinetics? Is the inflammation response stronger after even longer (48h, 72h) exposure to Cs-BMP?
- One main weak point of the study is that the conclusions of inflammatory activation via NF-κB and COX-2 are drawn from a single method (IF). There are couple of methods which can support observed results, e.g., a colorimetric NF-κB p65 transcription factor activation assay or Western blot analysis of COX-2. I completely understand the difficulty, which may occur implementing those methods in a setup where cells are exposed to Cs-BMP, especially involving analysis of the spatial distribution of inflammation. However, incorporating them might give a more general picture of Cs-BMP exposure effects on evoking inflammatory response, especially in comparison to more homogenous or non-uniform exposure to ionizing radiation.
- On Figure 6 the Authors reported that out-of-field cells in a non-uniform exposure to ionizing radiation are very radiosensitive, as per clonogenic survival data, and link this to NF-κB and COX-2 activation. Can this effect be rescued by NF-κB and/or COX-2 inhibition? Is it possible to make a more direct comparison between in- and out-of-field cells in a lower range of absorbed dose, rather than relying on extrapolated data between 0 and 1Gy?
- On Figure 4 the Authors report a relationship between inflammatory signaling (NF-κB and COX-2) and DNA damage (radiation induced γH2A.X foci), but data about γH2A.X foci appear to be taken from another publication (so the data is already published?). It would be more beneficial to investigate a possible correlation between DNA damage and inflammation signaling based on data coming from the same experiment, utilizing e.g., triple staining for involved factors.
- Additional control for referring intracellular signaling as a main source of the effects observed in distal cells would be very welcome. Can the effects in distal cells reported in this study be reproduced also by in-direct co-culturing or use of conditional media (from irradiated or proximal to Cs-BMP cells)?
Minor points:
- It is typical to set a significance level for a biological study to 5%, while everything above that value is considered as a trend or a tendency. The Authors decided to include also α=10% as significant results, however, to avoid confusion, I would suggest reporting p-values instead for results between 5-10% significant levels.
- Figure descriptions are missing the definition of error bars presented on the figures (S.E.M. or S.D.?)
- Are the Authors referring to a ‘scenario’ model? (lines 353 & 354)
- How was the normality of γH2A.X foci distribution tested? Were all data tested for normal distribution before choosing an appropriate statistical test?
Author Response
We would like to thank the reviewer #2 for careful reading and comments. We revised the manuscript based on the comments. Please see the attachment.

Reviewer 3 Report
Back in 2011, the tsunami disaster led to large amounts of radioactive materials emission at FDNPP. Cs-BMPs, found two years later, are insoluble radioactive microparticles, that were estimated to be widely existed surrounding FDNPP. Cs-BMPs are accessible to respiratory system and induce both acute and chronic local internal exposure and threaten the tissue integrity. It is thus crucial to understand the potential biological risk of Cs-BMPs exposure. Studies from Dr. Sato group and others reported the spatial distribution of DSBs upon local Cs-BMPs exposure. In which, local Cs-BMPs exposure results in more DNA damage to distal cells, but less DNA damage to proximal cells. The authors speculated that the communications between damaged cells and undamaged cells modulate the distinct DNA damage responses upon local Cs-BMPs exposure. The authors set up a well-controlled approach to investigate the spatial distribution of DSBs and inflammatory signaling for cells upon uniform or non-uniform exposure to Cs-BMPs and X-ray. Besides DSBs, the authors also probed the inflammatory responses and found dramatic NF-κB p65 and COX-2 expressions, which may be involved in regulating the spatial distribution of DSBs. Upon local Cs-BMPs exposure, more NF-κB p65 were found in proximal cells, while NF-κB p65 + and COX-2 + double positive cells were found more activated in distal cells, in contrast to 137Cs γ-ray.
Chronic exposure of insoluble radioactive microparticles brought up the concerns of biosafety in a long run. Radioactive materials act as DNA damage agents and threaten genome integrity. It is of great importance to understand the DNA damage response and its regulation when organisms are exposed. I have a few concerns that should be dealt with prior to publication.
Major concerns:
- The authors should better define the effects of chronic local Cs-BMPs exposure. In this regard, is 24hrs treatment sufficient for this statement? Long-term exposure is necessary to strengthen this statement. In a previous study, Harding et al. reported a chronic DNA damage induced inflammatory signaling at day 3~6 (or later) (Harding, et al. Nature, 2017).
- Representative immunofluorescence images of γ-H2AX foci, NF-κB p65 and COX-2 are necessary to help readers better understand the concepts of damage induced DSBs and inflammatory signaling.
- Giemsa-stained colonies are also necessary to better show clonogenic assay.
- In Figure 4, the authors reasoned that “ATM and NF-κB modulator cooperate to activate NF-κB pathway in response to DNA damage”, in both local Cs-BMPs exposure and uniform Cs-137 exposure. Since it is well known ATM activation is responsible for DNA damage response signaling when treated with Cs-137. So it is worthwhile to test whether ATM inhibitor impairs NF-κB p65 and COX-2 double positive inflammatory signaling.
- In Figure 5, the authors used nuclear “number of γ-H2AX foci per cell” to evaluate DNA damage, which shows a moderate difference between proximal and distal cells. Does the γ-H2AX foci intensity better support the statement?
- In Figure 5 (Line 279), the authors stated that “inhibition of NF-κB and COX-2 re- 279
duces DNA damage (DSB) induction in the cells at 1650–1950 μm”. However, only Figure 5F (HBEC3-KT) results support this argument.
Miner concerns:
- More description of NF-κB p65 and COX-2 mediated inflammatory responses is required to help readers better understand the reason why authors specifically looked at these two factors.
- Line 283: “figure 5B and 5E” should be “Figure 5B and 5E”. Line 297: “figure 5C and 5F” should be “Figure 5C and 5F”. And probably somewhere else.
- Line 357: Typo, “exposur” should be “exposure”.
Author Response
We would like to thank the reviewer #3 for careful reading. We revised the manuscript based on the comments. Please see the attachment.

Round 2
Reviewer 2 Report
I would like to thank the Authors for their efforts to answer my comments and suggestions attentively and extensively. The responses are considered adequate and convincing. Overall changes incorporated in the revised version clearly improved the manuscript and further underline significance of the work. Authors included also additional discussion points about future directions of the presented work, justifying enough not adding additional experiments in the current manuscript. To sum up, the manuscript can be accepted in the present form.